# Cognitive costs and gait parameters during single- and dual-task conditions: A comparative study in individuals with and without non-specific neck pain

Ibrahim M. Moustafa[1,2]*, Shorouk Abu-Ghosh[1], Amal Ahbouch[1,2]
Shima Abdollah Mohammad Zadeh[1,2], Meeyoung Kim[1,2], Iman Khowailed[1,2]

1 Department of Physiotherapy, College of Health Sciences, University of Sharjah, Sharjah, United Arab Emirates, 2 Neuromusculoskeletal Rehabilitation Research Group, RIMHS–Research Institute of Medical and Health Sciences, University of Sharjah, Sharjah, United Arab Emirates

* labuamr@sharjah.ac.ae

## Abstract

### Background

Non-specific neck pain (NSNP) is increasingly common among students, often due to long hours of sitting and frequent use of electronic devices. This widespread issue underscores the importance of understanding how NSNP affects students' cognitive abilities and motor functions. This study aims to evaluate the effects of NSNP on dual-task performance by analyzing the gait parameters and cognitive performance during both single-task and dual-task conditions.

### Methods

Forty-five participants with NSNP and forty-five age-matched controls were assessed using an optical motion-capture system. Participants underwent gait assessments during both single-task (without cognitive load) and dual-task conditions, in which the cognitive tasks involved simple mathematical computations.

### Results

Results revealed that under single-task conditions, differences in gait parameters between groups were not statistically significant. However, under dual-task conditions, participants with NSNP exhibited significant impairments in gait parameters and higher cognitive costs ($p < 0.05$). Correlation analysis indicated that pain intensity was significantly associated with cognitive cost and gait parameter alterations during dual-task conditions ($p < 0.05$).

**Data availability statement:** The data has been submitted as Supporting information.

**Funding:** The author(s) received no specific funding for this work.

**Competing interests:** The authors have declared that no competing interests exist.

## Conclusions

These findings suggest that NSNP significantly elevates the cognitive effort required during the dual-task. These findings emphasize the need for interventions to alleviate neck pain and improve both physical and cognitive health.

---

## 1. Introduction

Neck pain has become increasingly prevalent among students, mainly due to prolonged periods of studying, sedentary behavior, and extensive use of electronic devices [1,2]. As students spend more time hunched over books and screens, their necks bear the burden of the strain [3]. This non-specific neck pain (NSNP), which is not linked to any specific pathology or anatomical defect, affects a significant number of students and can lead to considerable discomfort, impacting not only their academic performance but also their overall quality of life [4].

The prevalence of neck pain among students highlights the importance of understanding its causes and effects. Factors such as poor posture, lack of ergonomic workstations, and inadequate physical activity contribute to the development of this condition [5]. Evidence indicates that improper use of pillows, prolonged use of electronic devices, and poor sitting posture significantly increase the risk of neck pain in this population [2]. Moreover, neck pain is influenced by various modifiable and non-modifiable risk factors, such as advanced age, female gender, low social support, and a history of neck or lower back pain, emphasizing the need for prevention and early diagnosis [1].

It is crucial for students to take proactive steps to alleviate neck pain, such as practicing proper posture, taking regular breaks, incorporating neck stretches into their daily routines, and seeking medical attention if necessary [6]. By managing neck pain and prioritizing their well-being, students can improve their focus, productivity, and overall health. Educational institutions can also contribute by promoting ergonomic practices and providing resources that support physical well-being [7].

While the physical symptoms of NSNP are well-documented, there is a significant and noteworthy surge in interest surrounding the comprehensive understanding of its cognitive implications. It is undeniably established that chronic pain, including neck pain, can profoundly and unequivocally influence cognitive function through an intricate network of mechanisms [8]. These mechanisms encompass pain-induced distraction, extensively altered sensory input, and the overwhelming psychological stress that is inherently intertwined with the chronically discomforting nature of this condition [9]. These interrelated factors collectively impose a substantial cognitive burden burden upon individuals, rendering the performance of tasks that necessitate concurrent physical and cognitive effort all the more arduous and formidable [10].

Literature has shown that chronic pain conditions can impair cognitive performance, particularly in tasks requiring attention, memory, and executive function [9]. The exploration of the relationship between neck pain and cognitive performance has garnered increasing attention in recent years, particularly in the context of student

populations [10,11]. However, the effects of mild to moderate neck pain on gait parameters under dual-task conditions are still not clear. The dual-task paradigm, which involves performing a cognitive task while simultaneously engaging in a physical activity, is particularly useful for studying the influence of cognitive load on motor performance [12].

Understanding dual-task performance provides essential insights into the impact of neck pain on individuals' ability to effectively manage multiple concurrent tasks [13]. This is crucial for various daily activities where cognitive-motor integration is essential [14]. For instance, students frequently find themselves needing to engage in conversations while walking or engaging in mental calculations while in motion. These situations necessitate seamless coordination between cognitive and motor functions [13]. However, for individuals experiencing neck pain, the additional burden of discomfort and pain may intensify the challenges associated with maintaining equilibrium and coordination, potentially resulting in compromised performance for both tasks at hand [15].

The purpose of this study is to investigate the effects of NSNP on dual-task performance by examining spatiotemporal gait parameters, including gait speed, stride length, and cadence, as well as cognitive performance under single-task and dual-task conditions among students. By comparing individuals with and without neck pain.

Given the increasing recognition of the cognitive implications of non-specific neck pain (NSNP), particularly in relation to motor-cognitive interactions, further exploration of its impact on dual-task performance is warranted. While previous research has examined the effects of chronic pain on cognitive function, the extent to which NSNP influences gait parameters and cognitive cost under dual-task conditions remains unclear. This study hypothesizes that individuals with NSNP will exhibit significantly higher cognitive costs during dual-task gait assessments compared to those without NSNP. Additionally, we hypothesize that NSNP will be associated with alterations in gait parameters, particularly under dual-task conditions, reflecting increased cognitive-motor interference. Furthermore, we predict that higher pain intensity will be positively correlated with greater cognitive costs and altered gait performance, supporting the premise that pain imposes an additional cognitive burden on task execution. By testing these hypotheses, this study aims to provide empirical evidence on the interplay between pain, cognitive load, and motor function, contributing to the development of targeted interventions that alleviate the cognitive burden associated with NSNP and enhance both physical and cognitive health outcomes.

## 2. Materials and methods

### 2.1. Study design, participants and setting

This comparative study was designed to assess the impact of cognitive dual-task on gait parameters in students experiencing NSNP, in comparison to a well-matched control group without neck pain. The research was conducted in the laboratories of the University of Sharjah, with subjects recruited from among the university's students through social media outreach. Recruitment took place from January to May 2024.

### 2.2. Ethical considerations

The study received ethical approval from the University of Sharjah's College of Health Sciences (Ethical approval number: REC-23-05-11-03-S). All participants provided their informed written consent prior to the start of data collection. The study adhered to established ethical guidelines and regulations.

### 2.3. Inclusion and exclusion criteria

• Participants were eligible for the NSNP group if they met the following criteria:

1. Age between 18 and 24 years;

2. Experience of NSNP for at least three months, with a pain intensity rating between 30 and 70 on the Visual Analog Scale (VAS);

3. Affiliation as a student at the University of Sharjah;

4. Willingness to participate in the study and provide informed consent.

    The control group consisted of individuals within the same age range and affiliation who reported no neck pain.

• Participants were excluded from the study if they had:

1. Previous fractures or operations on the neck or shoulders.

2. Previous major injuries or surgical procedures involving the musculoskeletal system.

3. Diagnosis of fibromyalgia, cervical radiculopathy/myelopathy, or cognitive impairments.

4. Deformities in the spine or limbs.

5. Participants with known vestibular disorders that could impact balance and gait were excluded.

6. Participants with recent history (within 6 months) of structured physical training targeting gait, balance, or cognitive-motor dual-tasking were excluded to control for training-related confounding factors.

7. These criteria ensured a homogeneous sample, reducing potential confounding factors that could influence gait or cognitive performance outcomes.

## 2.3. Study tools and outcome measures

**2.3.1. Visual analog scale.** Participants quantified their neck pain intensity using a visual analog scale, a validated and reliable instrument for measuring pain intensity.

**2.3.2. BTS GAITLAB system.** The spatiotemporal and kinematic parameters of gait were assessed using an optical motion-capture system consisting of 8 infrared cameras (Smart-D, BTS Bioengineering, Milan, Italy) running at a frequency of 120 Hz. Anthropometric measurements, such as height, weight, and various body segment widths, were obtained before the experimental tests. Spherical reflective passive markers were placed on the subjects' skin following the protocol outlined by Davis et al.

**2.3.3. Dual Task Cost (DTC) percentage.** The analysis was based on average performance from three dual-task trials to establish a comprehensive measure of the impact of cognitive load on gait. To calculate the Dual Task Cost (DTC) percentage, the disparity between single and dual-task performance scores was determined and then divided by the single-task performance score before being multiplied by 100. For instance, if the average speed during single-task walking was 1.2 m/s and during dual-task walking was 1.0 m/s, the DTC percentage would be calculated as $((1.2–1.0)/\ 1.2) \times 100 = 16.67\%$.

**2.3.4. Spatiotemporal parameters.** The spatiotemporal parameters of the participants, including step length, speed, and cadence, were measured under both single-task and dual-task conditions using the BTS gait lab system.

## 2.4. Study procedure

Participants were asked to walk at their chosen pace along a 10-meter walkway, with their 3D marker trajectories recorded by cameras under two different scenarios: single-task and dual-task. In the single-task scenario, participants walked at their chosen speed without performing any additional tasks, completing three trials to establish their baseline gait characteristics. In the dual-task scenario, participants walked while engaging in various cognitive tasks, such as answering yes or no questions, counting, or performing simple math calculations. Each participant completed a minimum of three trials for each scenario and were given rest periods in between the trials to reduce the effect of the fatigue. Upon completion of the tests, the raw data were analyzed using specialized software (Smart Analyzer, BTS Bioengineering, Milan, Italy) to derive the necessary parameters.

**2.4.1. Cognitive task design and standardization.** To evaluate the effects of cognitive-motor dual-task, participants were required to perform mental arithmetic tasks while walking. The mathematical computations were carefully designed to impose a consistent cognitive load across all participants while ensuring feasibility in a dynamic setting. The specific details of the cognitive task design are as follows:

*Task Description:*

• Participants were presented with auditory arithmetic problems that involved addition, subtraction, and multiplication of single- and double-digit numbers (e.g., *23 + 17, 45–19, 8 × 7*).

• These problems were delivered via a standardized voice recording played through speakers in the testing room to maintain uniform timing across trials.

*Task Difficulty Standardization:*

• The difficulty level was moderate, ensuring that participants were cognitively engaged but not overwhelmed.

• Problems were chosen based on prior dual-task research, where similar arithmetic operations have been validated to generate measurable cognitive-motor interference [12].

• Each trial consisted of five sequential arithmetic problems, and participants had to provide verbal responses while walking.

*Adaptation to Individual Cognitive Ability:*

• To prevent ceiling effects **(task too easy) or** floor effects **(task too difficult), a** pre-trial cognitive screening was performed using a brief 5-item arithmetic test.

• Participants who solved four or more problems correctly within 5 seconds were assigned more complex problems (e.g., *two-digit multiplication*).

• Those who answered fewer than two problems correctly received simpler problems (e.g., *single-digit operations*).

• This adaptation ensured that the cognitive task imposed a comparable cognitive load across all participants.

*Response Measurement and Recording:*

• Verbal responses were recorded using a high-sensitivity microphone, and both response accuracy (%) and response time (s) were documented for each problem.

• Response time was measured from the moment the problem was presented until the participant provided an answer.

• If a participant failed to answer within 6 seconds, the problem was marked as "missed."

## 2.5. Sample size determination

The sample size for this study was determined to ensure adequate statistical power to detect clinically meaningful differences between participants with non-specific neck pain (NSNP) and controls under dual-task conditions. Given the importance of distinguishing not only statistically significant but also functionally relevant differences, our sample size estimation incorporated effect size benchmarks aligned with prior gait and cognitive-motor interference research.

Based on a review of relevant literature and preliminary data from comparable studies, an effect size (Cohen's d) of 0.5 was selected as an appropriate threshold for identifying moderate but clinically significant differences in gait parameters and cognitive costs associated with NSNP. This effect size aligns with previous studies investigating dual-task interference in pain populations [10,16].

To ensure methodological rigor, a power analysis was conducted using G*Power 3.1 software. The following parameters were applied:

- Effect size (Cohen's d) = 0.5 (moderate clinical effect)

- Significance level (α) = 0.05 (two-tailed)

- Desired statistical power (1 − β) = 0.80

- Allocation ratio = 1:1 (equal group sizes for NSNP and controls)

The results of this power analysis indicated that a minimum of 40 participants per group would be required to detect a significant difference between groups with 80% power at a 5% significance level. To account for potential dropouts and missing data, we increased the sample size by an additional 5 participants per group, leading to a final sample size of 45 participants per group (total N = 90).

## 2.6. Statistical analysis

To assess the normal distribution of all baseline descriptive variables, the Kolmogorov-Smirnov test was utilized. Continuous variables are presented as mean values alongside their standard deviations (SD). Levene's test was applied at a 95% confidence interval to examine the homogeneity of variances, with significance defined as p-values less than 0.05. Descriptive statistics (mean ± SD) were reported for each respective time point. To confirm group equivalence, chi-squared tests were applied to categorical variables, while Student's t-tests were used for continuous variables. The comparison of mean values between groups was conducted using the Student's t-test, with statistical significance established at a p-value of less than 0.05. Effect size was assessed using Cohen's d, where d values around 0.2, 0.5, and 0.8 corresponded to negligible, moderate, and substantial clinical significance, respectively. Pearson's correlation coefficients (r) were utilized to investigate the associations between neck pain severity and spatiotemporal gait parameters in both single and dual-task conditions, as well as the relationship between neck pain severity and cognitive cost. All analyses were performed using SPSS version 20.0 (SPSS Inc., Chicago, IL, USA), ensuring the assumptions of normality and homogeneity of variance were met prior to analysis.

## 3. Results

### 3.1. Demographics

Over 200 potential participants were initially screened, with the most common exclusion criterion being the pain levels outside 30–70 on the VAS or having neck pain for less than three months. Ultimately, 45 participants with VAS (25 males, 20 females) and 45 matched controls (based on age, BMI, and sex) without neck pain were recruited for the study (p > 0.05). The significant difference in VAS validates the classification of the groups based on neck pain status. The demographic characteristics and clinical variables of these participants are detailed in Table 1.

### 3.2. Single task detailed results

Table 2 presents the mean differences in spatiotemporal parameters between the neck pain and no neck pain groups during the single-task condition. Most variables exhibit non-significant differences. Most spatiotemporal parameters during the single-task condition do not show significant differences between the neck pain and no neck pain groups. However, significant differences were observed in swing time, mean velocity and step length, indicating an impact of neck pain on these specific parameters. Fig 1 illustrates the comparison of Single-Task Gait Parameters.

### 3.3. Dual task detailed results

Table 3 illustrates the mean differences in spatiotemporal parameters between the neck pain and no neck pain groups during the dual-task condition. Significant differences were observed in all spatiotemporal parameters, except for the

**Table 1. Baseline demographic characteristics and clinical variables.**

| Variable | Neck Pain (n = 45) | No Neck Pain (n = 45) | p-value | Cohen's d |
|---|---|---|---|---|
| Age (years) | 20.50 ± 2.0 | 20.00 ± 3.0 | 0.30 | 0.67 |
| BMI | 21.50 ± 1.7 | 21.40 ± 1.8 | 0.60 | 0.60 |
| Gender (% Male) | 55.5 | 55.5 | 1.00 | < 0.001 |
| Gender (% Female) | 44.5 | 44.5 | 1.00 | < 0.001 |
| VAS | 55 ± 8 | 0 | | |
| General Health Status (scale 1–5) | 4.8 ± 0.8 | 4.7 ± 0.5 | 0.47 | 0.15 |
| Daily Routine (hours of sedentary behavior) | 6.5 ± 2 | 7 ± 1.5 | 0.18 | −0.28 |

**Table 2. Mean Differences between Spatiotemporal Parameters in Neck Pain and No Neck Pain Groups during Single Task.**

| Variable | Group | Mean ± SD | 95% CI | Cohen's d | p-value |
|---|---|---|---|---|---|
| Stride time (s) | Neck Pain | 1.25 ± 0.10 | [1.22, 1.28] | 0.20 | 0.15 |
| | No Neck Pain | 1.23 ± 0.10 | [1.20, 1.26] | | |
| Stance time (s) | Neck Pain | 0.75 ± 0.05 | [0.73, 0.77] | 0.25 | 0.10 |
| | No Neck Pain | 0.74 ± 0.05 | [0.72, 0.76] | | |
| Swing time (s) | Neck Pain | 0.45 ± 0.05 | [0.43, 0.47] | 0.60 | < 0.001 |
| | No Neck Pain | 0.42 ± 0.05 | [0.40, 0.44] | | |
| Stance phase (%) | Neck Pain | 62.0 ± 2.0 | [61.3, 62.7] | 0.30 | 0.05 |
| | No Neck Pain | 61.5 ± 2.0 | [60.8, 62.2] | | |
| Double support phase (%) | Neck Pain | 12.0 ± 1.5 | [11.5, 12.5] | 0.15 | 0.20 |
| | No Neck Pain | 11.8 ± 1.5 | [11.3, 12.3] | | |
| Mean velocity m/s | Neck Pain | 1.10 ± 0.10 | [1.07, 1.13] | 0.40 | 0.02 |
| | No Neck Pain | 1.15 ± 0.10 | [1.12, 1.18] | | |
| Mean velocity height (%height/s) | Neck Pain | 70.0 ± 5.0 | [68.5, 71.5] | 0.10 | 0.50 |
| | No Neck Pain | 69.8 ± 5.0 | [68.3, 71.3] | | |
| Cadence (steps/min) | Neck Pain | 110.0 ± 5.0 | [108.5, 111.5] | 0.20 | 0.15 |
| | No Neck Pain | 109.0 ± 5.0 | [107.5, 110.5] | | |
| Stride length (%height) | Neck Pain | 75.0 ± 5.0 | [73.5, 76.5] | 0.15 | 0.20 |
| | No Neck Pain | 74.5 ± 5.0 | [73.0, 76.0] | | |
| Step length (m) | Neck Pain | 0.60 ± 0.05 | [0.58, 0.62] | 0.60 | < 0.001 |
| | No Neck Pain | 0.57 ± 0.05 | [0.55, 0.59] | | |
| Step width avg (m) | Neck Pain | 0.09 ± 0.03 | [0.08, 0.10] | 0.10 | 0.50 |
| | No Neck Pain | 0.09 ± 0.03 | [0.08, 0.10] | | |

cadence, between the neck pain and no neck pain groups during the dual-task condition. This suggests that neck pain significantly affects gait parameters under increased cognitive load. Fig 2 shows the comparison of Single-Task Gait Parameters.

### 3.4. Cognitive cost detailed results

The cognitive cost comparison results, presented in Table 4, show significant differences across multiple gait parameters in the neck pain group. The Mean Difference in Cognitive Cost (CC) between Neck Pain and No Neck Pain. The cognitive cost values indicate that individuals with neck pain incur significantly higher cognitive costs across various gait parameters during dual-task conditions compared to those without neck pain. This finding suggests that neck pain exacerbates the cognitive effort required to maintain gait.

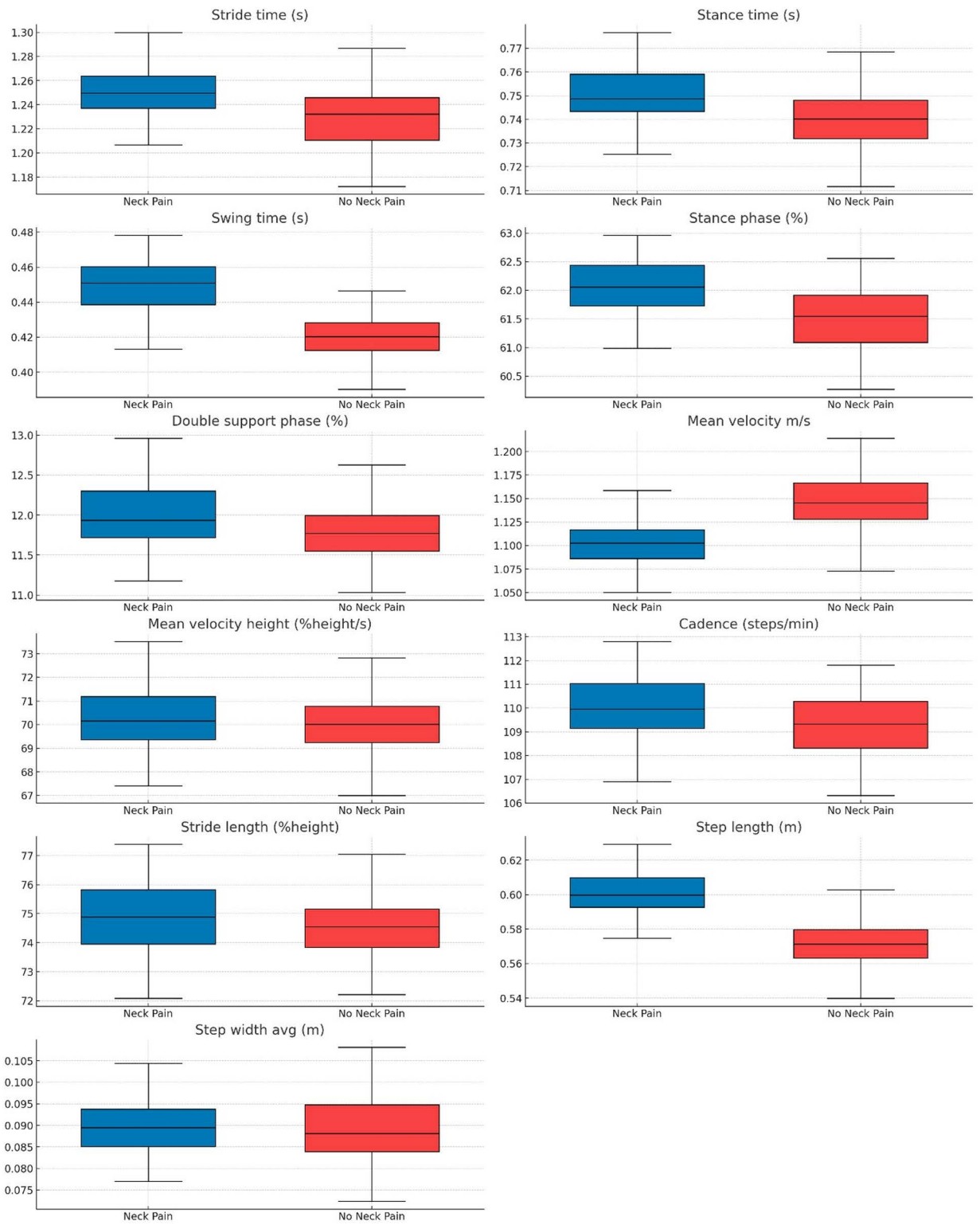

**Fig 1. Box Plot Comparison of Single-Task Gait Parameters.**

**Table 3. Mean Differences in Spatiotemporal Parameters Between Neck Pain and No Neck Pain Groups During Dual-Task Conditions.**

| Variable | Group | Mean±SD | 95% CI | Cohen's d | p-value |
|---|---|---|---|---|---|
| Stride time (s) | Neck Pain | 1.50±0.15 | [1.45, 1.55] | 0.80 | < 0.001 |
| | No Neck Pain | 1.20±0.10 | [1.17, 1.23] | | |
| Stance time (s) | Neck Pain | 0.80±0.05 | [0.78, 0.82] | 1.14 | < 0.001 |
| | No Neck Pain | 0.70±0.05 | [0.68, 0.72] | | |
| Swing time (s) | Neck Pain | 0.50±0.05 | [0.48, 0.52] | 0.80 | < 0.001 |
| | No Neck Pain | 0.45±0.05 | [0.43, 0.47] | | |
| Stance phase (%) | Neck Pain | 65.0±3.0 | [64.1, 65.9] | 0.80 | 0.01 |
| | No Neck Pain | 61.0±2.5 | [60.3, 61.7] | | |
| Double support phase (%) | Neck Pain | 13.0±2.0 | [12.4, 13.6] | 0.80 | < 0.001 |
| | No Neck Pain | 11.0±1.5 | [10.5, 11.5] | | |
| Mean velocity m/s | Neck Pain | 1.10±0.10 | [1.07, 1.13] | 0.80 | < 0.001 |
| | No Neck Pain | 1.20±0.10 | [1.17, 1.23] | | |
| Mean velocity height (%height/s) | Neck Pain | 65.0±5.0 | [63.5, 66.5] | 0.80 | < 0.001 |
| | No Neck Pain | 72.0±5.0 | [70.5, 73.5] | | |
| Cadence (steps/min) | Neck Pain | 110.0±5.0 | [108.5, 111.5] | 0.60 | 0.30 |
| | No Neck Pain | 105.0±5.0 | [103.5, 106.5] | | |
| Stride length (%height) | Neck Pain | 75.0±5.0 | [73.5, 76.5] | 1.00 | < 0.001 |
| | No Neck Pain | 80.0±5.0 | [78.5, 81.5] | | |
| Step length (m) | Neck Pain | 0.60±0.05 | [0.58, 0.62] | 1.10 | < 0.001 |
| | No Neck Pain | 0.65±0.05 | [0.63, 0.67] | | |
| Step width avg (m) | Neck Pain | 0.09±0.03 | [0.08, 0.10] | 0.60 | < 0.001 |
| | No Neck Pain | 0.07±0.02 | [0.06, 0.08] | | |

### 3.5. Correlational results

The single-task condition reveals no significant correlations between pain intensity and gait parameters (p > 0.05). This indicates that pain intensity does not significantly influence performance parameters during single tasks in either group.

The correlation table for the dual-task condition illustrates significant correlations between pain intensity and various performance parameters, suggesting that pain intensity substantially influences gait parameters during dual-task activities (Table 5). The heatmap visually represents the relationships between pain intensity and gait parameters (Fig 3).

The correlation table for cognitive cost values illustrates a connection between pain intensity and cognitive cost measures (Table 6). The results suggest that pain intensity significantly influences the cognitive cost related to various gait parameters, indicating that as pain intensity rises, the cognitive effort needed to sustain gait during dual-task scenarios also increases. The heatmap visually represents the relationship between pain intensity (VAS) and cognitive cost (Fig 4).

### 3.6. Cognitive task performance

To assess the cognitive workload imposed by the arithmetic task, response accuracy (%) and response time (s) were analyzed. Participants with NSNP demonstrated significantly lower accuracy (p < 0.001) and longer response times (p < 0.001) during dual-task conditions compared to controls. These findings confirm heightened cognitive cost in the NSNP group during dual-task execution. The results are summarized in Table 7.

## 4. Discussion

Our study underscores the profound impact of non-specific neck pain (NSNP) on cognitive performance during dual-task activities, highlighting the intricate relationship between musculoskeletal discomfort and cognitive function. The findings

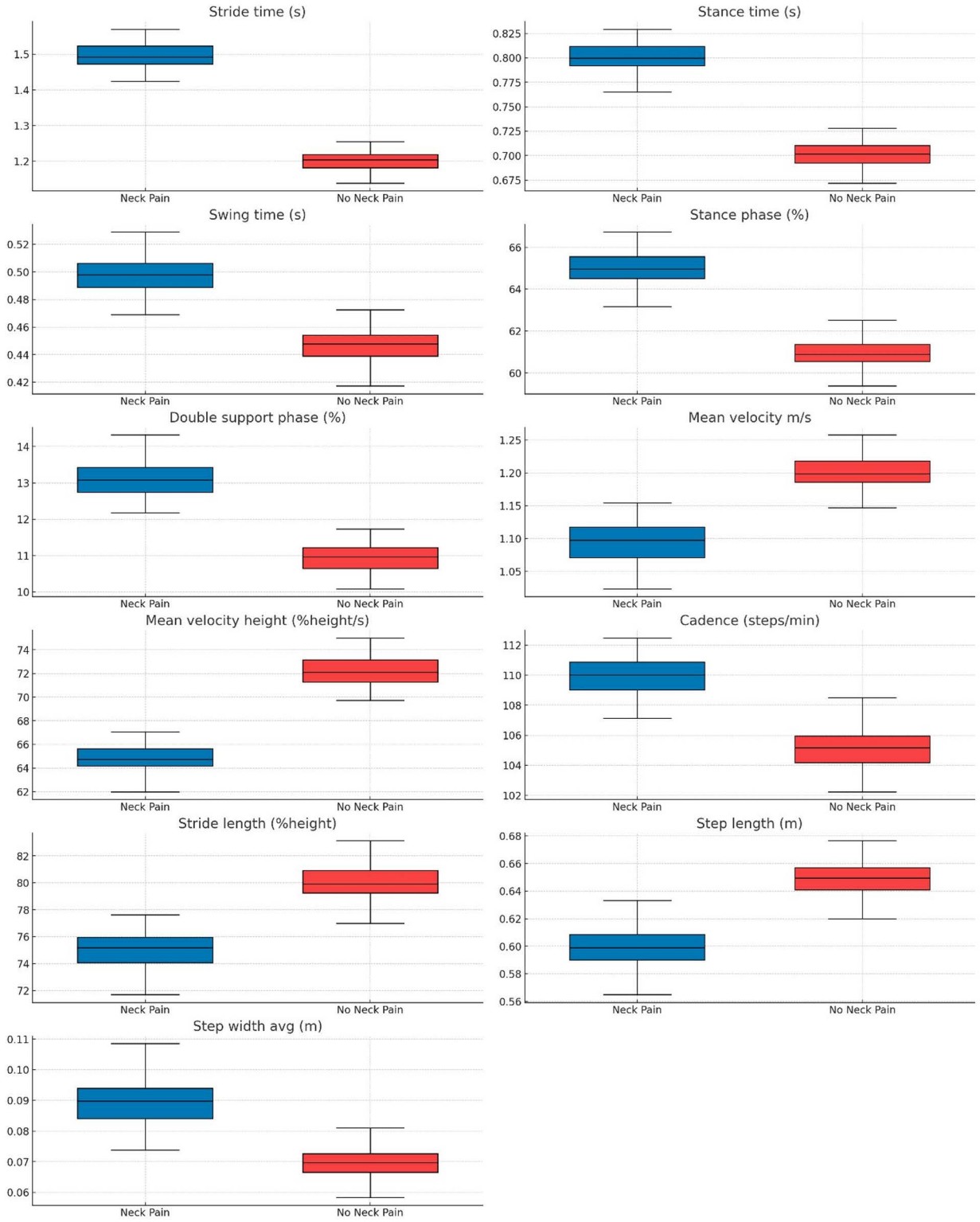

**Fig 2. Box Plot Comparison of dual-Task Gait Parameters.**

**Table 4. Mean Differences in cognitive cost Between Neck Pain and No Neck Pain groups.**

| Variable | Group | Mean±SD | 95% CI | Cohen's d | p-value |
|---|---|---|---|---|---|
| CC Stride time | Neck Pain | 25.79±5.95 | [23.99, 27.59] | 0.87 | < 0.001 |
| | No Neck Pain | 5.19±3.50 | [4.19, 6.19] | | |
| CC Stance time | Neck Pain | 8.43±4.00 | [7.43, 9.43] | 0.89 | < 0.001 |
| | No Neck Pain | -6.75±3.75 | [-7.75, -5.75] | | |
| CC Swing time | Neck Pain | 2.78±3.00 | [2.08, 3.48] | -0.72 | < 0.001 |
| | No Neck Pain | 15.03±4.68 | [13.53, 16.53] | | |
| CC Stance phase | Neck Pain | 8.49±3.19 | [7.49, 9.49] | 0.64 | 0.01 |
| | No Neck Pain | 1.40±3.00 | [0.70, 2.10] | | |
| CC Double support phase | Neck Pain | 19.87±5.69 | [18.07, 21.67] | 0.66 | 0.01 |
| | No Neck Pain | 1.11±5.00 | [-0.39, 2.61] | | |
| CC Mean velocity | Neck Pain | 12.36±6.45 | [10.36, 14.36] | -0.75 | 0.02 |
| | No Neck Pain | 6.85±4.00 | [5.85, 7.85] | | |
| CC Mean velocity height | Neck Pain | 13.34±4.90 | [11.84, 14.84] | -0.80 | < 0.001 |
| | No Neck Pain | 3.18±4.00 | [2.18, 4.18] | | |
| CC Cadence | Neck Pain | 4.49±4.00 | [3.49, 5.49] | -0.30 | < 0.001 |
| | No Neck Pain | 1.46±4.00 | [0.46, 2.46] | | |
| CC Stride length | Neck Pain | 6.84±4.00 | [5.84, 7.84] | -0.81 | 0.01 |
| | No Neck Pain | 2.41±4.00 | [1.41, 3.41] | | |
| CC Step length | Neck Pain | 9.98±5.00 | [8.48, 11.48] | -1.13 | 0.01 |
| | No Neck Pain | 4.39±4.00 | [3.39, 5.39] | | |
| CC Step width avg | Neck Pain | 12.43±4.90 | [10.93, 13.93] | 0.49 | 0.01 |
| | No Neck Pain | 9.91±5.00 | [8.41, 11.41] | | |

**Table 5. Correlation Between Pain Intensity and Gait Parameters During Dual-Task Conditions.**

| Parameter | Spearman r | p-value |
|---|---|---|
| Stride time (s) | -0.68 | < 0.001 |
| Stance Time (s) | -0.56 | < 0.001 |
| Swing Time (s) | 0.31 | < 0.001 |
| Stance Phase (%) | -0.56 | < 0.001 |
| Swing Phase (%) | 0.53 | < 0.001 |
| Single Support Phase (%) | 0.40 | < 0.001 |
| Double Support Phase (%) | -0.55 | < 0.001 |
| Mean Velocity (m/s) | 0.28 | < 0.001 |
| Mean Velocity (%height/s) | 0.49 | < 0.001 |
| Cadence (steps/min) | 0.20 | < 0.001 |
| Stride Length (%height) | 0.39 | < 0.001 |
| Step Length (m) | 0.60 | < 0.001 |
| Step Width (m) | -0.40 | < 0.001 |

reveal that individuals with NSNP experience significantly higher cognitive costs when performing dual-task gait assessments, reinforcing the idea that chronic neck pain imposes additional cognitive demands on individuals managing both motor and cognitive tasks simultaneously.

These observations align with previous research highlighting the cognitive demands imposed by chronic pain [8–10], suggesting that addressing these cognitive demands is crucial for improving functional outcomes in patients with NSNP.

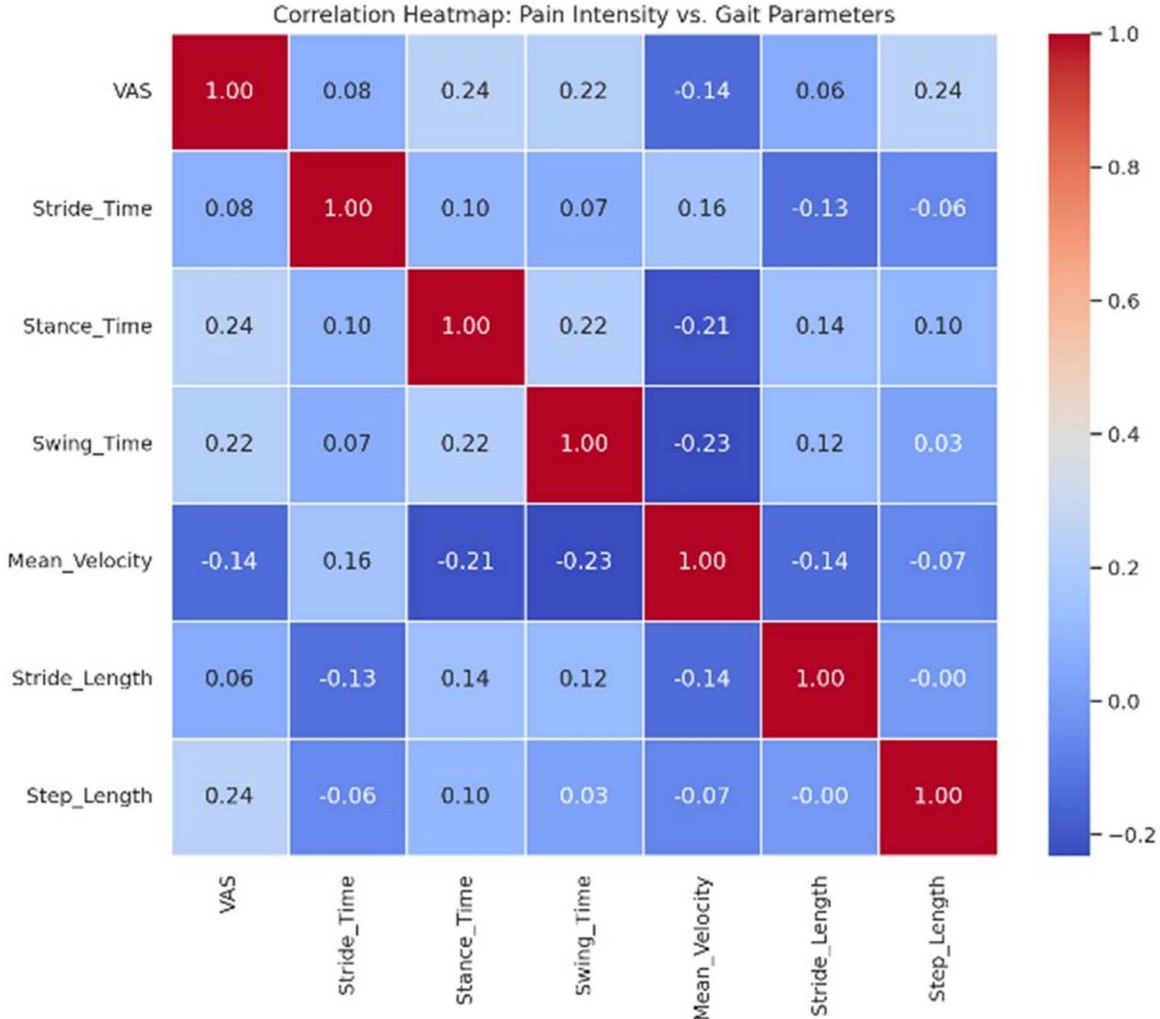

**Fig 3. The heatmap visually represents the relationship between pain intensity (VAS) and gait parameters.**

However, our findings also indicate that gait performance in single-task conditions remains largely unaffected, suggesting a more complex compensatory mechanism at play.

### 4.1. Compensatory mechanisms in single-task vs. dual-task conditions

Despite the significant cognitive impact observed under dual-task conditions, our results show no statistically significant differences in most gait parameters between NSNP and control groups under single-task conditions. However, this does not necessarily imply the absence of an underlying impairment. It is possible that individuals with NSNP engage in subtle compensatory mechanisms, allowing them to maintain near-normal gait patterns when cognitive demands are low.

These compensatory strategies likely stem from altered proprioceptive feedback, a common consequence of chronic musculoskeletal pain, particularly in the cervical region. Research suggests that NSNP patients rely more heavily on visual and vestibular inputs to maintain postural stability, compensating for their impaired proprioception [17,18]. However, these mechanisms become insufficient under dual-task conditions, where increased cognitive demand competes for attentional resources, leading to measurable declines in gait performance.

**Table 6. Correlation Between Pain Intensity and Cognitive Cost.**

| Parameter | Spearman r | p-value |
|---|---|---|
| CC Stride time (s) | -0.61 | < 0.001 |
| CC Stance Time (s) | -0.41 | < 0.001 |
| CC Swing Time (s) | -0.35 | < 0.001 |
| CC Stance Phase (%) | -0.35 | < 0.001 |
| CC Swing Phase (%) | -0.45 | < 0.001 |
| CC Single Support Phase (%) | -0.23 | 0.04 |
| CC Double Support Phase (%) | -0.21 | 0.04 |
| CC Mean Velocity (m/s) | -0.37 | 0.01 |
| CC Mean Velocity (%height/s) | -0.30 | 0.01 |
| CC Cadence (steps/min) | -0.22 | 0.01 |
| CC Stride Length (%height) | -0.32 | 0.01 |
| CC Step Length (m) | -0.55 | < 0.001 |
| CC Step Width (m) | -0.50 | < 0.001 |

CC: Cognitive cost.

This intricate interplay between sensorimotor function and cognition highlights the importance of future research incorporating kinematic analyses, electromyography (EMG), and center-of-mass tracking. These techniques could help uncover neuromuscular adaptations facilitating compensation and inform targeted rehabilitation strategies to optimize both cognitive and motor function in individuals with NSNP.

### 4.2. Neurobiological mechanisms underlying cognitive and motor impairments in NSNP

The neurobiological mechanisms underlying the observed impairments in cognitive and motor functions among individuals with NSNP can be attributed to altered sensorimotor integration, proprioceptive deficits, and cortical reorganization. Chronic neck pain disrupts afferent input from the cervical region, leading to maladaptive changes in the sensorimotor cortex and impairments in postural control [18]. These proprioceptive deficits, primarily due to cervical dysfunction, affect head and neck position sense, increasing reliance on visual and vestibular cues, which in turn heightens cognitive load during motor tasks [17]. Additionally, pain-induced changes in functional connectivity within the prefrontal cortex, a critical area for attention and executive control, compound these effects [9]. This intricate cascade of changes underscores the need for addressing sensorimotor dysfunction and cognitive-motor interference when designing rehabilitation strategies for individuals with NSNP.

### 4.3. The impact of pain on cognitive and motor control

The impact of pain on cognitive and motor control is well documented, especially in contexts where sensory-motor feedback is compromised due to musculoskeletal conditions. Pain alters cortical processing and proprioception, increasing the sensorimotor effort required for stability [17]. The heightened cognitive effort needed for postural control in NSNP suggests that cognitive overload may lead to diminished dual-task performance, a finding supported by recent studies [19,20]. Moreover, chronic pain has been linked to neuroplastic changes in attentional control networks, underscoring the role of cognitive load in pain management [9]. By elucidating these complex interactions, our study contributes to a deeper understanding of NSNP and supports the integration of cognitive load theory into pain management strategies.

The discrepancies in dual-task performance observed between the NSNP and control groups resonate with findings from prior studies that suggest musculoskeletal discomfort significantly influences cognitive processes [13,15,21]. Notably, the elevated cognitive load in the NSNP group can be linked to disrupted sensory input and proprioception due to pain

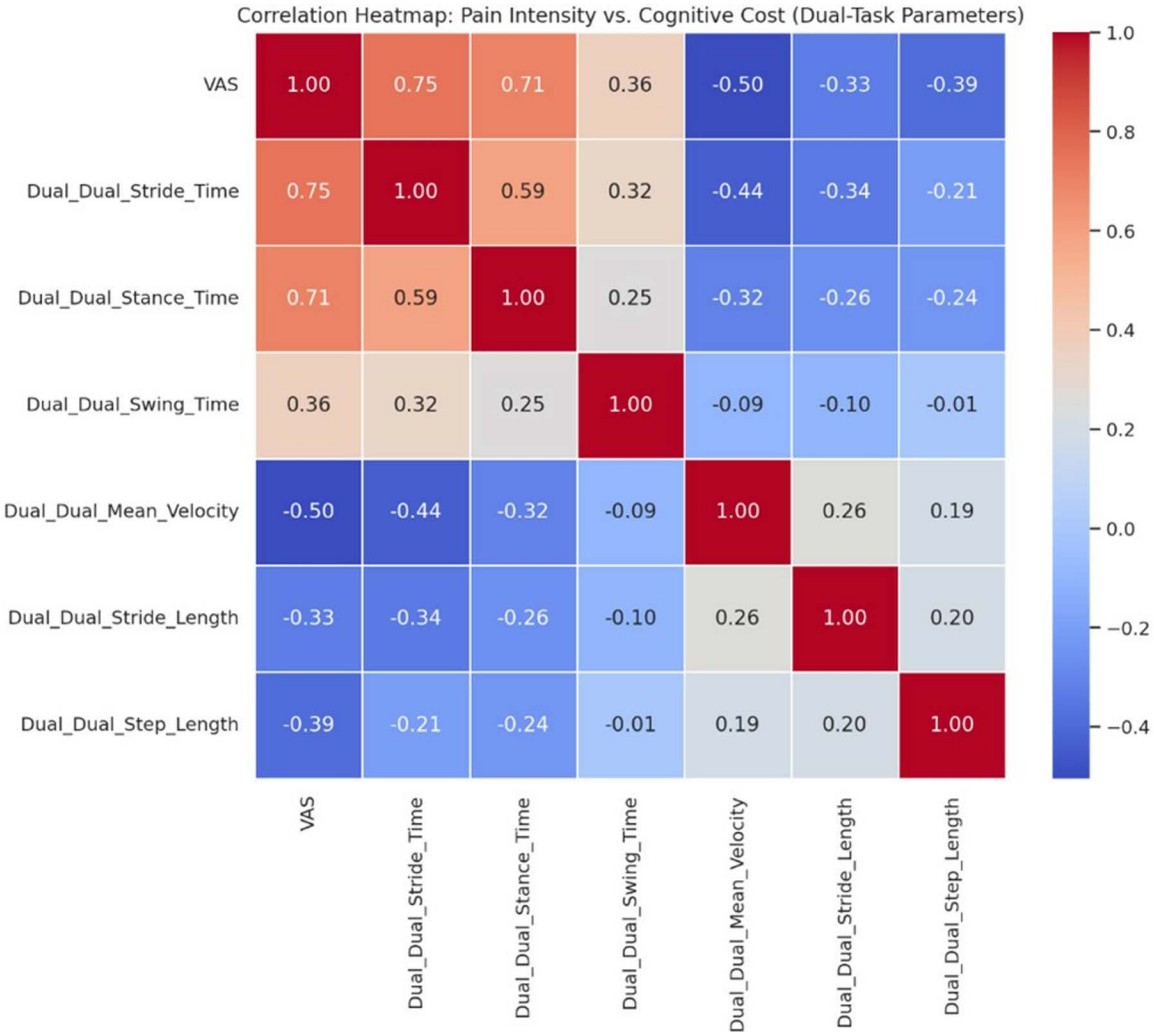

**Fig 4.** **The heatmap visually represents the relationship between pain intensity (VAS) and cognitive cost.**

**Table 7. Cognitive Task Performance.**

| Condition | Neck Pain Group (Mean±SD) | No Neck Pain Group (Mean±SD) | p-value | Cohen's d |
|---|---|---|---|---|
| Single-Task Accuracy (%) | 89.2±5.6 | 91.3±4.9 | 0.18 | 0.42 |
| Dual-Task Accuracy (%) | 73.4±8.9 | 85.7±6.2 | <0.001 | 0.86 |
| Single-Task Response Time (s) | 3.1±0.8 | 2.9±0.7 | 0.22 | 0.30 |
| Dual-Task Response Time (s) | 4.7±1.1 | 3.8±0.9 | <0.001 | 0.91 |

[18], closely mirroring phenomena such as phantom limb pain where maladaptive cortical reorganization follows sensory input loss. This suggests that similar neuroplastic changes may occur in NSNP, affecting the sensory and motor cortices and leading to chronic pain [8,9]. Such neuroplastic alterations result in significant cognitive burdens as the brain compensates for impaired postural stability, manifesting as deficits in attention, processing speed, and memory in those with musculoskeletal pain [10].

NSNP impacts both motor and cognitive function, especially under dual-task demands. Understanding these effects through cognitive load theory and neurophysiological is key to guiding effective rehabilitation.

### 4.4. The role of cognitive load theory in understanding NSNP deficits

The complexities of cognitive-motor interactions in NSNP are further exemplified in dual-task performance, where the integration of cognitive and motor demands is crucial. The observed cognitive costs align with established research on dual-task paradigms, which emphasize the interplay between motor control and cognitive processing in daily activities [22].

From a cognitive load theory (CLT) perspective, our findings suggest that NSNP exacerbates cognitive demands required for task execution. According to Sweller et al. (2011), when intrinsic and extraneous cognitive loads exceed an individual's processing capacity, performance in one or both tasks declines [23]. This study highlights that pain increases extraneous cognitive load, impairing gait-related task efficiency. These findings advocate for targeted interventions aimed at reducing cognitive burdens, potentially improving both cognitive and motor function simultaneously.

### 4.5. The neurophysiological mechanisms

While our results demonstrate that individuals with non-specific neck pain (NSNP) exhibit increased cognitive costs and altered gait patterns under dual-task conditions, the underlying neurophysiological mechanisms require deeper examination. Chronic pain is known to disrupt prefrontal cortex (PFC) activity, leading to impairments in executive functions such as attention control, working memory, and decision-making [8,9]. Functional neuroimaging studies further indicate that chronic pain alters connectivity between the PFC, anterior cingulate cortex (ACC), and pain-processing regions, contributing to higher cognitive load and attentional interference [24]. This neural disruption may explain the greater cognitive effort required by individuals with NSNP when performing motor-cognitive dual-tasks.

Crucially, neuroplasticity plays a key role in the interplay between cognitive processes and musculoskeletal discomfort. Neuroplastic adaptations resulting from altered sensory input due to neck pain can have both positive and negative cognitive outcomes. Studies indicate that neuroplastic mechanisms, such as synaptogenesis and cortical reorganization, are engaged to accommodate the elevated cognitive demands imposed by chronic pain [9]. These processes highlight the brain's remarkable ability to reorganize and recalibrate its neural networks in response to altered sensory inputs, preserving cognitive function despite physical impairments.

This substantial cognitive burden imposed by chronic pain stems from the continuous activation of pain-related neural networks, which interfere with working memory and executive function [9,25]. The dual-task paradigm is particularly relevant in this context, as both cognitive and motor functions rely on overlapping prefrontal regions [22]. The heightened cognitive costs observed in individuals with NSNP can be attributed to the increased attentional demand required to override pain-induced cognitive interference [26]. Furthermore, pain-related neuroplasticity in the PFC and sensorimotor cortex can lead to maladaptive cortical reorganization, further compromising cognitive performance [27]. These findings highlight the importance of integrating cognitive training interventions alongside physical therapy to address both motor and cognitive impairments in individuals with chronic pain.

### 4.6. Implications for clinical practice

The robust correlations between pain intensity and cognitive dual-task performance emphasize the necessity for comprehensive assessment protocols that address both musculoskeletal and cognitive dimensions. Clinicians should recognize the potential cognitive effects of neck pain and incorporate strategies to manage musculoskeletal discomfort within their rehabilitation approaches [28]. This is particularly crucial for populations at higher risk of cognitive decline, including older adults and individuals with neurological disorders [29,30]. Our research also highlights the importance of dual-task gait assessments as a crucial tool for identifying cognitive impairments. By integrating cognitive and motor tasks, these assessments provide a more realistic measure of everyday functioning, making them essential for detecting subtle

cognitive deficits that may not be evident in single-task evaluations [31]. These findings have significant implications for early intervention and continuous monitoring of cognitive health in clinical practice. Clinicians should incorporate neck pain assessments into routine evaluations, especially for individuals exhibiting cognitive concerns [32]. Embracing interdisciplinary approaches involving physical therapists, occupational therapists, and cognitive specialists can provide comprehensive care that addresses both the physical and cognitive challenges associated with neck pain [33]. Early intervention and patient education emphasizing the importance of proper posture and effective management of musculoskeletal discomfort are essential for mitigating the risk of developing neck pain and its consequent cognitive impairments [34].

## 4.7. Practical applications

Given our findings that individuals with NSNP exhibit significant cognitive costs during dual-task performance, traditional physical therapy interventions focused solely on musculoskeletal rehabilitation may be insufficient. Evidence suggests that chronic musculoskeletal pain disrupts cognitive function, particularly executive processes such as attention and working memory [8,9]. As such, an integrated rehabilitation approach that combines musculoskeletal therapy with cognitive training is recommended. Cognitive-motor training, which involves dual-task exercises designed to improve cognitive resource allocation while performing motor tasks, may help individuals with NSNP mitigate cognitive interference and enhance motor control [22]. Neuroplasticity-based approaches, including motor imagery training and task-oriented exercises, may further optimize sensorimotor integration and prefrontal cortex engagement [24]. Our study underscores the need for integrating cognitive evaluations into the standard clinical assessment of patients with chronic pain, as recommended by earlier study [35]. By doing so, treatment plans can be better tailored to address both the physical and cognitive aspects of NSNP, potentially improving overall treatment outcomes.

## 4.8. Limitations and future recommendations

While this study provides valuable insights, several limitations must be acknowledged. First, its cross-sectional design prevents establishing causal relationships between non-specific neck pain (NSNP) and cognitive-motor performance. While strong associations were observed between pain intensity, cognitive cost, and gait alterations, it remains unclear whether these impairments are a consequence of NSNP or if individuals with pre-existing cognitive deficits are more prone to developing NSNP. Longitudinal studies are needed to clarify this relationship.

Second, the study sample consisted exclusively of young university students, limiting generalizability to older adults and individuals with chronic musculoskeletal conditions. Given age-related declines in cognitive-motor integration and proprioception [13,22], these effects may be even more pronounced in aging populations. Future research should assess whether rehabilitation strategies should be tailored based on age-related sensory changes.

Third, we did not conduct a gender-based analysis, despite evidence that women may experience greater pain sensitivity and cognitive costs in dual-task conditions due to hormonal and neuroplastic differences [36,37]. Future studies should include gender-stratified analyses to determine whether intervention strategies need to be sex-specific.

Fourth, pain variability was not systematically recorded, although participants were instructed to report significant pain changes. While no participants reported increased pain during or after gait assessments, a more structured pain monitoring approach would strengthen future research.

Fifth, further exploration of the neurophysiological mechanisms linking musculoskeletal discomfort and cognitive function is necessary. Examining brain connectivity changes in individuals with NSNP could enhance our understanding of sensorimotor and executive function disruptions. Additionally, intervention studies assessing the impact of pain relief on cognitive outcomes could provide evidence for multimodal rehabilitation approaches, integrating physical therapy, cognitive training, and pain management strategies.

Finally, fatigue effects were not systematically assessed, though participants were given rest periods between trials. Physical and cognitive fatigue may have accumulated over successive attempts, potentially affecting gait performance

and cognitive costs. Given that muscle fatigue and cognitive depletion can impair neuromuscular control and executive function [10,38], future research should incorporate objective fatigue assessments (e.g., EMG monitoring, reaction time tracking, or subjective fatigue scales) and implement counterbalanced testing designs to control for fatigue-related performance declines.

By addressing these limitations, future research can enhance the clinical applicability of dual-task interventions and refine rehabilitation strategies for individuals with NSNP.

## 5. Conclusions

This study demonstrates that individuals with non-specific neck pain (NSNP) exhibit increased cognitive demands during dual-task walking, leading to significant gait alterations and higher cognitive costs compared to pain-free controls. These findings suggest that NSNP impairs cognitive-motor integration, potentially compromising postural stability and attentional control during divided-attention tasks.

Given these associations, further research is needed to determine whether these impairments progress over time or can be effectively managed through intervention. Longitudinal studies should examine the long-term impact of NSNP on cognitive-motor function, while intervention research should prioritize dual-task training, sensorimotor rehabilitation, and cognitive therapy to develop targeted treatment strategies. Addressing these challenges will help refine rehabilitation approaches, ensuring better mobility, cognitive efficiency, and overall functional outcomes for individuals with NSNP.

## Supporting information

**S1 Data. Raw data.**
(XLSX)

## Author contributions

**Conceptualization:** Ibrahim M. Moustafa.

**Data curation:** Shorouk Abu-Ghosh, Shima Abdollah Mohammad Zadeh, Iman Khowailed, Meeyoung Kim.

**Formal analysis:** Ibrahim M. Moustafa, Amal Ahbouch, Iman Khowailed.

**Investigation:** Shorouk Abu-Ghosh, Amal Ahbouch, Shima Abdollah Mohammad Zadeh, Iman Khowailed, Meeyoung Kim.

**Methodology:** Shorouk Abu-Ghosh, Amal Ahbouch, Shima Abdollah Mohammad Zadeh, Meeyoung Kim.

**Project administration:** Ibrahim M. Moustafa.

**Supervision:** Ibrahim M. Moustafa, Iman Khowailed, Meeyoung Kim.

**Validation:** Ibrahim M. Moustafa, Iman Khowailed.

**Visualization:** Amal Ahbouch.

**Writing – original draft:** Ibrahim M. Moustafa, Shorouk Abu-Ghosh, Amal Ahbouch, Shima Abdollah Mohammad Zadeh, Iman Khowailed, Meeyoung Kim.

**Writing – review & editing:** Ibrahim M. Moustafa, Shorouk Abu-Ghosh, Amal Ahbouch, Shima Abdollah Mohammad Zadeh, Iman Khowailed, Meeyoung Kim.

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
