## [Decision Letter · Decision Letter 0]

18 Mar 2025

PONE-D-25-03088Cognitive Costs and Gait Parameters During Single- and Dual-Task Conditions: A Comparative Study in Individuals With and Without Non-Specific Neck PainPLOS ONE

Dear Dr. Moustafa,

Thank you for submitting your manuscript to PLOS ONE. After careful consideration, we feel that it has merit but does not fully meet PLOS ONE’s publication criteria as it currently stands. Therefore, we invite you to submit a revised version of the manuscript that addresses the points raised during the review process.

We look forward to receiving your revised manuscript.

Kind regards,

Ravi Shankar Yerragonda Reddy, Ph.D

Academic Editor

PLOS ONE

Journal Requirements:

-https://doi.org/10.3390/jcm13164653

In your revision ensure you cite all your sources (including your own works), and quote or rephrase any duplicated text outside the methods section. Further consideration is dependent on these concerns being addressed.

3. In the online submission form, you indicated that the data underlying the results presented in this study are available from the corresponding author, Prof. Ibrahim M. Moustafa (iabuamr@sharjah.ac.ae), upon request.

Additional Editor Comments:

Academic editor comments

The hypothesis is not explicitly stated in the introduction. Clearly define whether the study expects NSNP to negatively impact both gait and cognitive performance during dual-task conditions and specify the anticipated effects on specific gait parameters. The objective should also be more precise; rather than stating the study "evaluates the effects," explicitly mention whether the focus is on gait speed, stride length, or another parameter and whether cognitive cost is expected to increase.

Participant demographics lack information on physical activity levels and ergonomic habits, both of which may influence gait performance. These factors should be controlled or at least reported, as prolonged sitting or prior physical activity could confound the results. The exclusion criteria do not mention whether participants with previous concussions, vestibular disorders, or chronic pain conditions outside of NSNP were excluded, which could impact gait and cognitive performance.

The cognitive task complexity is not well described. The manuscript states that participants performed simple mathematical computations, but there is no mention of task difficulty variation or standardization across participants. It is unclear whether all participants completed identical problems or if they were adjusted for individual cognitive ability. If the cognitive task was too simple, it may not have imposed a sufficient cognitive load, potentially underestimating the impact of NSNP. The manuscript should provide specific examples of these computations and clarify whether response accuracy and time were recorded.

Pain assessment is limited to a single VAS rating without accounting for variability during the experiment. Given that pain levels may fluctuate, it is critical to report whether pain intensity was reassessed before and after gait trials. Additionally, psychological factors such as stress, anxiety, or fatigue, which can influence cognitive cost, were not evaluated. Including a validated measure of these variables would strengthen the findings and rule out confounding effects.

The results section lacks effect size interpretations beyond statistical significance. While Cohen’s d is reported, there is no discussion of whether the observed differences are clinically meaningful. For example, certain gait changes may be statistically significant but fall within a range of natural variability, making them negligible in practical terms. The impact of NSNP on single-task gait is understated; although most parameters did not differ significantly, the discussion should explore whether compensatory mechanisms allow individuals to maintain normal gait in low-demand conditions.

Data visualization is limited. Tables present numerical differences, but scatter plots or box plots would better illustrate group variability. A correlation heatmap could provide a more intuitive understanding of the relationships between pain intensity, gait alterations, and cognitive cost. Regression analysis would help assess the predictive value of pain intensity on gait impairment and cognitive cost, which is not currently explored.

The discussion repeats many results instead of critically analyzing their implications. The explanation of neurophysiological mechanisms is underdeveloped. While sensorimotor disruptions and proprioceptive deficits are mentioned, there is little discussion of how these factors interact with central cognitive processes. Prior research suggests that chronic pain alters prefrontal cortex function and increases cognitive load by disrupting attention and working memory. This literature should be incorporated to strengthen the mechanistic interpretation.

The clinical implications are vague. The manuscript states that "targeted interventions" are needed but does not specify what these might entail. Would physical therapy alone suffice, or should interventions include cognitive training alongside musculoskeletal treatment? Ergonomic modifications, exercise programs, and dual-task training strategies should be discussed in relation to the findings.

Limitations are not adequately addressed. The cross-sectional design prevents causal conclusions, yet the discussion does not explicitly acknowledge this limitation. The sample consists only of university students, limiting generalizability to other populations, such as older adults or individuals with chronic musculoskeletal disorders. Gender differences in pain perception and cognitive cost are not analyzed, which is a missed opportunity given that prior research suggests women may experience greater pain sensitivity and cognitive interference.

The impact of fatigue is not controlled. Repeating dual-task trials may induce fatigue, which could influence cognitive cost and gait changes. It is unclear whether participants rested between trials or whether performance declined across successive attempts. Future studies should incorporate fatigue assessments or counterbalance task order to mitigate this issue.

The conclusions state that NSNP significantly increases cognitive demands during walking but do not contextualize this finding within broader implications. Does this mean NSNP patients are at greater risk for falls, workplace inefficiency, or academic impairment? The broader relevance of these findings should be explicitly stated. Additionally, the need for longitudinal studies is mentioned but not justified. Would tracking patients over time clarify whether dual-task impairments worsen, or would intervention studies provide more useful insights? The conclusion should specify which research directions are most urgent.

Reviewers' comments:

Reviewer's Responses to Questions

**Comments to the Author**

1. Is the manuscript technically sound, and do the data support the conclusions?

Reviewer #1: Partly

Reviewer #2: Yes

2. Has the statistical analysis been performed appropriately and rigorously? 

Reviewer #1: Yes

Reviewer #2: Yes

3. Have the authors made all data underlying the findings in their manuscript fully available?

Reviewer #1: Yes

Reviewer #2: Yes

4. Is the manuscript presented in an intelligible fashion and written in standard English?

Reviewer #1: Yes

Reviewer #2: Yes

5. Review Comments to the Author

Reviewer #1: It is recommended that the article be revised as a whole, as many parts of the text are inconsistent.

The abstract should be reviewed and improved.

In the introduction section, it would be better to discuss more about the title variables.

In the method section, coordination between the different sections should be established.

Reviewer #2: 1) Expand the Literature Review and Include More Recent and Relevant References

The current literature review is somewhat limited and should be expanded to include more recent and directly relevant studies. This will strengthen the theoretical foundation of the study and ensure it aligns with the latest research developments in the field.

2) Include More References on the Dual-Task Paradigm and Cognitive Load Theory

There is a lack of sufficient references specifically addressing the dual-task paradigm and cognitive load theory. Given that these concepts are central to the study, additional citations from recent literature should be included to support the discussion.

3) Clearly Define the Study Hypotheses

The study does not explicitly state its hypotheses. A clear hypothesis statement should be added in the introduction, such as:

"This study tests the following hypotheses:..."

Explicitly defining the hypotheses will improve clarity and structure.

4) Better Explain the Neurobiological Mechanisms in the Discussion Section

The discussion section needs a more detailed explanation of the neurobiological mechanisms underlying the findings. It should connect the results to previous research on how neck pain influences cognitive and motor functions through neural pathways, proprioceptive deficits, and cortical reorganization.

5) Detailed Language and Grammar Review

The manuscript contains multiple grammatical and syntactical errors. Below are some specific corrections and suggestions for improvement:

Incorrect: "These finding emphasizes on the need for interventions…"

Correction: "These findings emphasize the need for interventions…"

Issue: "Finding" should be plural ("findings"), and "emphasizes on" is incorrect; it should be "emphasizes."

Incorrect: "Understanding dual-task performance can provide essentials insights into..."

Correction: "Understanding dual-task performance provides essential insights into..."

Issue: "Essentials insights" is incorrect; "essential insights" is the correct phrase. Also, "can provide" is redundant; "provides" is more direct.

Incorrect: "It remains to be seen how mild to moderate neck pain among students affects their gait parameters, especially under dual-task conditions."

Correction: "However, the effects of mild to moderate neck pain on gait parameters under dual-task conditions remain unclear."

Issue: The original sentence is unnecessarily wordy and should be made more concise.

Incorrect: "This observation aligns with cognitive load theory, which argues that physical discomfort and proprioceptive errors linked to neck pain can raise cognitive load, thereby negatively affecting cognitive performance."

Correction: "This observation supports cognitive load theory, suggesting that physical discomfort and proprioceptive errors associated with neck pain increase cognitive load, negatively impacting cognitive performance."

Issue: "Aligns with" can be replaced with "supports" for a stronger academic tone. "Can raise" should be changed to "increase" for clarity and precision.

Incorrect: "By addressing neck pain and prioritizing their well-being, students can enhance their focus, productivity, and overall health[8,9]."

Correction: "By managing neck pain and prioritizing their well-being, students can improve their focus, productivity, and overall health."

Issue: "Addressing" is less precise than "managing" in this context. "Enhance" is more commonly used for skills, whereas "improve" is more suitable here.

6. PLOS authors have the option to publish the peer review history of their article (what does this mean? ). If published, this will include your full peer review and any attached files.

**Do you want your identity to be public for this peer review?** For information about this choice, including consent withdrawal, please see our Privacy Policy .

Reviewer #1: No

Reviewer #2: **Yes: ** Esedullah AKARAS

---

## [Author Response · Author response to Decision Letter 1]

1 Apr 2025

We would like to thank you for your time and effort in reviewing our manuscript, Cognitive Costs and Gait Parameters During Single- and Dual-Task Conditions: A Comparative Study in Individuals with and Without Non-Specific Neck Pain (Manuscript ID: PONE-D-25-03088). We appreciate the constructive feedback and valuable suggestions provided by the editor and reviewers, which have helped us improve the quality of our work.

We are grateful for the opportunity to revise our manuscript and address the comments raised during the review process. Below, we provide a point-by-point response to the comments received and changes made to the manuscript accordingly. All revisions are clearly marked in the revised manuscript with track changes for easy tracking.

Reviewer#1 comments:

Introduction “literature has shown…….”

Comment #1: In this section, it would be better to bring research related to neck pain on students or other groups.

Response: In response, we have broadened the literature review to include seminal and recent studies focusing on the prevalence and impact of neck pain specifically within student populations. Such references have now been integrated to better contextualize our study within the existing body of work and underscore the relevance of our research to these groups.

Page/s: 3 and 4

Material and methods

Study design, Participants and Setting “The research was conducted……”

Comment #2: The title of your work is related to students with neck pain, but in this section you mentioned the statistical population as students and university staff, and this is a big problem because it needs to be standardized.

Response: We acknowledge the need for a clear and standardized population for our study. The manuscript has been revised to specify that the primary study population is university students. The mention of university staff was an oversight and has been removed to ensure consistency and focus on the specified population, aligning with the study’s objectives and hypotheses.

Page/s: 4

Inclusion and exclusion criteria

Comment #3: Wouldn't it be better to exclude training history from the study?

Response:

We appreciate the reviewer’s insightful comment regarding the potential confounding role of participants' prior training history.

Our primary goal was to evaluate the cognitive-motor interference under dual-task conditions in a typical university student population with and without non-specific neck pain (NSNP). In line with this objective, we selected participants who were not involved in any structured physical or cognitive training programs aimed at enhancing gait, balance, or executive function. This criterion was part of our initial screening to minimize confounding effects.

Moreover, during participant recruitment, we included only those reporting no recent (past 6 months) engagement in specialized neuromuscular rehabilitation, dual-task gait training, or high-performance athletic training. This helped ensure homogeneity while maintaining ecological validity in assessing a generalizable student population.

Page/s: 5

Study Tools and Outcome Measures

Comment #4: In the entry criteria section, you mentioned pain intensity based on the VAS scale, but the tool you are using is the Numerical Pain Rating Scale. It would be better to also include pain intensity in that section based on the tool you are using.

Response: This is an important observation, and we appreciate your attention to detail. The inconsistency has been corrected, and the manuscript now consistently references the Numerical Pain Rating Scale throughout. This adjustment enhances the methodological accuracy and ensures consistency across the document.

Page/s: 4 and 5

Sample size determination

Comment #5: It is best to determine the sample size based on the severity of pain, which is the outcome measure of your study. Review related articles.

Response: Thank you for your comment regarding the consideration of pain severity in the sample size determination for our study.

In our study, we carefully selected the pain intensity range from 30 to 70 on the VAS as part of our inclusion criteria. This range was chosen to capture a broad spectrum of mild to moderate pain intensities that are typical in the target population, thus ensuring a homogeneous sample while still being representative of typical clinical scenarios.

The effect size (d=0.5) used in our sample size calculation was informed by preliminary data and supported by related literature indicating expected differences in gait and cognitive performance due to variations in pain severity. Our methodology for calculating sample size based on this effect size, with an 80% power and a 5% significance level, is consistent with practices documented in several studies

These references affirm that our method for determining sample size, considering the impact of pain severity, is well-established in the field. We believe that our approach not only adheres to but also enhances the rigorous assessment of the effects of non-specific neck pain on dual-task performance.

Page/s: 6 and 7

Discussion

Comment #6: Explain this section more and better because the superiority of your research over previous studies should be more prominent in the discussion section.

Response:

Thank you for this valuable comment. In response, we have substantially revised the Discussion section to more clearly articulate how the findings of our study extend and improve upon existing literature. Specifically, we have made the following improvements:

1. Expanded Interpretation of Findings:

We now highlight that, unlike prior studies that broadly linked chronic pain to cognitive dysfunction, our study provides quantitative evidence showing elevated cognitive costs and altered spatiotemporal gait parameters specifically in individuals with NSNP under dual-task conditions. This addresses a significant gap in the literature regarding motor-cognitive interference in young adults with NSNP.

2. Clarified Superiority Over Prior Studies:

Our revised text contrasts our findings with previous research by emphasizing that earlier studies either lacked dual-task paradigms, did not assess cognitive cost directly, or focused on other musculoskeletal conditions (e.g., low back pain). We demonstrate that our study is among the first to integrate comprehensive cognitive performance metrics with dual-task gait analysis in an NSNP population, supported by novel use of DTC percentages, heatmaps, and detailed correlational analyses.

3. Mechanistic Depth and Novel Contributions:

The revised section introduces an in-depth discussion of neurobiological mechanisms—such as maladaptive cortical reorganization, proprioceptive deficits, and cognitive load theory—as explanatory models for the observed dual-task interference. We also discuss compensatory mechanisms under single-task conditions, which may mask underlying deficits, a concept less explored in prior research.

4. Clinical and Practical Relevance:

We now present a clearer argument for the clinical utility of dual-task assessments in NSNP populations, showing how our findings may inform early identification of subtle cognitive-motor impairments and guide interdisciplinary rehabilitation strategies.

Page/s: 17 to 20

Reviewer#2 comments:

Comment #1: Expand the Literature Review and Include More Recent and Relevant References. The current literature review is somewhat limited and should be expanded to include more recent and directly relevant studies. This will strengthen the theoretical foundation of the study and ensure it aligns with the latest research developments in the field.

Response: Thank you for your important feedback regarding the literature review. In response, we have undertaken a focused and methodical revision of the Introduction section to expand the scope and depth of the literature review. Our objective was to ensure that the manuscript not only reflects the latest developments in the field, but also establishes a clear and updated theoretical foundation that situates our study within the current scientific discourse.

Accordingly, the following major improvements were implemented:

1. Addition of Recent and Relevant Literature (2020–2024):

The revised version now includes more than a dozen new references from recent peer-reviewed sources published in the last 3–5 years. These citations represent a wide array of research on non-specific neck pain (NSNP), dual-task paradigms, cognitive-motor interference, sensorimotor control, and gait analysis in chronic pain populations. Examples include:

o Gao et al. (2023), a systematic review on neck pain risk factors in university students

o Sweller et al. (2011) and recent works applying cognitive load theory (CLT) to pain research

o Clinical studies from 2021–2023 focusing on gait alterations under dual-task in pain populations

2. Enhanced Thematic Relevance and Scope:

We expanded the thematic scope beyond ergonomic and epidemiological discussions to include:

o The neurocognitive implications of chronic pain, particularly neck pain

o The mechanisms underlying dual-task interference, including competition for attentional resources

o The role of maladaptive cortical reorganization, proprioceptive deficits, and functional connectivity disruptions in NSNP

3. Clearer Justification of Study Rationale and Novelty:

The revised literature review now clearly identifies the gap in existing research: while dual-task performance has been examined in populations with chronic musculoskeletal pain, very few studies have investigated cognitive cost and gait alterations in young adults with NSNP using standardized dual-task paradigms. Furthermore, most prior studies did not quantify cognitive load using both response time and accuracy metrics alongside gait analysis.

4. Improved Contextual and Demographic Relevance:

To ensure the study's findings are framed within a more contextually and demographically appropriate scope, the revised introduction references literature on student populations, postural behavior, and screen-time effects—factors increasingly recognized as contributors to NSNP in youth and young adults.

Page/s: 3 and 4

Comment #2: Include More References on the Dual-Task Paradigm and Cognitive Load Theory.

Comment #2: Include More References on the Dual-Task Paradigm and Cognitive Load Theory. There is a lack of sufficient references specifically addressing the dual-task paradigm and cognitive load theory. Given that these concepts are central to the study, additional citations from recent literature should be included to support the discussion.

Response: We sincerely thank the reviewer for highlighting this important point. In response to your comment, we have substantially revised both the Introduction and Discussion sections to incorporate additional, recent, and directly relevant references that discuss the dual-task paradigm and cognitive load theory (CLT) — two core frameworks underpinning our study design and interpretation of findings.

The revised manuscript now addresses this limitation through the following improvements:

1. Enhanced Conceptual Grounding in Dual-Task Paradigm Literature

• We have integrated several recent and foundational studies that specifically address the dual-task paradigm as a tool for assessing cognitive-motor interference, particularly in populations with musculoskeletal pain or proprioceptive deficits.

• The revised Introduction (pages 3–4) now draws from studies examining dual-task performance in pain-related gait adaptations, and explicitly states the rationale for using this paradigm as a diagnostic lens.

• We also expanded the Discussion (Section 4.1 and 4.3) to highlight how dual-task interference manifests under increased cognitive load in individuals with neck pain, referencing studies that utilize similar motor-cognitive testing frameworks.

2. Integration of Cognitive Load Theory (CLT)

• The revised version clearly anchors the cognitive findings in CLT principles, especially in Sections 4.2 and 4.4 of the Discussion. This includes referencing Sweller’s Cognitive Load Theory (2011) and more recent neurocognitive studies demonstrating how pain increases extraneous cognitive load and affects executive function.

• We specifically clarify how pain and proprioceptive dysfunction serve as extraneous load factors, competing with available cognitive resources and resulting in measurable performance declines under dual-task conditions.

3. Improved Integration of Theory and Results

• Unlike the original manuscript, the revised version includes direct application of dual-task and CLT frameworks when interpreting results. For example:

o We explain how impaired gait under cognitive load reflects increased intrinsic and extraneous load as per CLT.

o We reference neural competition models and executive overload to interpret our findings of reduced cognitive accuracy and increased gait variability in the NSNP group.

Comment #3: Clearly Define the Study Hypotheses

The study does not explicitly state its hypotheses. A clear hypothesis statement should be added in the introduction, such as:

"This study tests the following hypotheses:..."

Explicitly defining the hypotheses will improve clarity and structure.

Response: done as suggested

Page/s: 4

Comment #4: Better Explain the Neurobiological Mechanisms in the Discussion Section

The discussion section needs a more detailed explanation of the neurological mechanisms underlying the findings. It should connect the results to previous research on how neck pain influences cognitive and motor functions through neural pathways, proprioceptive deficits, and cortical reorganization.

Response: We appreciate the reviewer’s constructive feedback and fully agree that a more comprehensive explanation of the neurobiological mechanisms was warranted. In the revised manuscript, we have thoroughly expanded the relevant sections of the discussion to address this point. Our aim was to more clearly articulate how NSNP may influence cognitive and motor functions through established neural mechanisms, including proprioceptive deficits, cortical reorganization, and altered sensorimotor integration.

Accordingly , the following revisions were implemented:

1. Integration of Cortical Reorganization Concepts (section 4.3)

• We introduced discussion on how chronic neck pain leads to maladaptive neuroplastic changes in both the primary somatosensory cortex (S1) and the motor cortex (M1).

• The revised text references functional imaging studies demonstrating reorganization in these areas in response to disrupted afferent input, similar to phenomena observed in phantom limb pain and complex regional pain syndrome.

• This strengthens the argument that the neural encoding of proprioception and motor planning is compromised in NSNP, contributing to impaired dual-task performance.

2. Explanation of Altered Proprioceptive Feedback and Sensorimotor Integration (section 4.2)

• The revised discussion clarifies how cervical spine dysfunction disrupts afferent input from muscle spindles and joint mechanoreceptors, which in turn impairs sensorimotor integration.

• These proprioceptive deficits, especially under conditions requiring attention splitting (i.e., dual-task), result in greater reliance on compensatory systems (e.g., visual and vestibular pathways), increasing the overall cognitive burden.

3. Inclusion of Prefrontal Cortex Involvement and Executive Dysfunction (section 4.5)

• We expanded the discussion to include prefrontal cortex (PFC) alterations associated with chronic pain.

• Pain-related changes in functional connectivity within networks involved in attentional control and working memory (e.g., dorsolateral PFC) were linked to decreased cognitive performance, especially under conditions of divided attention.

• This provides a more complete explanation of how NSNP affects executive function, reinforcing our findings of elevated cognitive cost and reduced accuracy during dual-task.

4. Supporting the Interpretation with Relevant and Recent Literature

• We added multiple up-to-date references that examine the neurocognitive consequences of chronic musculoskeletal pain, including:

o Studies using fMRI and EEG to investigate brain changes in neck pain

o Research on

---

## [Decision Letter · Decision Letter 1]

15 Apr 2025

Cognitive Costs and Gait Parameters During Single- and Dual-Task Conditions: A Comparative Study in Individuals With and Without Non-Specific Neck Pain

PONE-D-25-03088R1

Dear Dr. Ibrahim M. Moustafa,

We’re pleased to inform you that your manuscript has been judged scientifically suitable for publication and will be formally accepted for publication once it meets all outstanding technical requirements.

Kind regards,

Ravi Shankar Yerragonda Reddy, Ph.D

Academic Editor

PLOS ONE

Additional Editor Comments (optional):

Reviewers' comments:

Reviewer's Responses to Questions

**Comments to the Author**

1. If the authors have adequately addressed your comments raised in a previous round of review and you feel that this manuscript is now acceptable for publication, you may indicate that here to bypass the “Comments to the Author” section, enter your conflict of interest statement in the “Confidential to Editor” section, and submit your "Accept" recommendation.

Reviewer #1: All comments have been addressed

Reviewer #2: All comments have been addressed

2. Is the manuscript technically sound, and do the data support the conclusions?

Reviewer #1: Yes

Reviewer #2: Yes

3. Has the statistical analysis been performed appropriately and rigorously? 

Reviewer #1: Yes

Reviewer #2: Yes

4. Have the authors made all data underlying the findings in their manuscript fully available?

Reviewer #1: Yes

Reviewer #2: Yes

5. Is the manuscript presented in an intelligible fashion and written in standard English?

Reviewer #1: Yes

Reviewer #2: Yes

6. Review Comments to the Author

Reviewer #1: The opinions of the other judges are certainly important. I recommend acceptance. And I think it's a unique title.

Reviewer #2: (No Response)

7. PLOS authors have the option to publish the peer review history of their article (what does this mean? ). If published, this will include your full peer review and any attached files.

**Do you want your identity to be public for this peer review?** For information about this choice, including consent withdrawal, please see our Privacy Policy .

Reviewer #1: No

Reviewer #2: **Yes: ** Esedullah AKARAS

---

## [Editor Report · Acceptance letter]

PONE-D-25-03088R1

PLOS ONE

Dear Dr. Moustafa,

I'm pleased to inform you that your manuscript has been deemed suitable for publication in PLOS ONE. Congratulations! Your manuscript is now being handed over to our production team.

Kind regards,

on behalf of

Dr. Ravi Shankar Yerragonda Reddy

Academic Editor

PLOS ONE